# SCRaPL: A Bayesian hierarchical framework for detecting technical associates in single cell multiomics data

**Christos Maniatis**[1]*, **Catalina A. Vallejos**[2,3]*, **Guido Sanguinetti**[1,4]*

**1** School of Informatics, The University of Edinburgh, Edinburgh, United Kingdom, **2** The Alan Turing Institute, London, United Kingdom, **3** MRC Human Genetics Unit, Institute of Genetics and Cancer, Western General Hospital, The University of Edinburgh, Edinburgh, United Kingdom, **4** International School for Advanced Studies (SISSA-ISA), Trieste, Italy

\* s1315538@sms.ed.ac.uk (CM); catalina.vallejos@ed.ac.uk (CAV); gsanguin@sissa.it (GS)

**Data Availability Statement:** The mESC dataset is available in the Gene Expression Omnibus under accession number GSE121708 (https://www.ncbi.nlm.nih.gov/geo/query/acc.cgi?acc=GSE121708).

## Abstract

Single-cell multi-omics assays offer unprecedented opportunities to explore epigenetic regulation at cellular level. However, high levels of technical noise and data sparsity frequently lead to a lack of statistical power in correlative analyses, identifying very few, if any, significant associations between different molecular layers. Here we propose SCRaPL, a novel computational tool that increases power by carefully modelling noise in the experimental systems. We show on real and simulated multi-omics single-cell data sets that SCRaPL achieves higher sensitivity and better robustness in identifying correlations, while maintaining a similar level of false positives as standard analyses based on Pearson and Spearman correlation.

## Author summary

Single-cell multi-omics assays offer unprecedented opportunities to explore epigenetic regulation at cellular level. However, high levels of noise frequently hide genomics regions with strong epigenetic regulation or produce misleading results. By carefully addressing this common problem SCRaPL aims become a useful tool in the hands of practitioners seeking to understand the role of particular genomic regions in the epigenetic landscape. Using different single cell multi-omics datasets, we have demonstrated that SCRaPL can increase detection rates up to five times compared to standard practices. This can improve performance of tools used for post experimental analysis, but more importantly it can indicate currently unknown genomic regions worth to further investigate.

This is a *PLOS Computational Biology* Methods paper.

Similarly the mEBC dataset can be found in the official 10X Genomics website (https://www.10xgenomics.com/resources/datasets/fresh-embryonic-e-18-mouse-brain-5-k-1-standard-2-0-0). The PBMC dataset is also found in 10X Genomics website (https://support.10xgenomics.com/single-cell-multiome-atac-gex/datasets/1.0.0/pbmc_granulocyte_sorted_10k). A Python implementation of SCRaPL along with preprocessing scripts are available at https://github.com/chrmaniatis/SCRaPL.

**Funding:** Funding from Engineering and Physical Sciences Research Council (EPSRC) Centre for Doctoral Training in Data Science (grant EP/L016427/1) supported CM. The funders had no role in study design, data collection and analysis, decisions to publish, or preparation of the manuscript.

**Competing interests:** The authors have declared that no competing interests exist.

## Introduction

High throughput single cell assays based on next generation sequencing are revolutionising our understanding of biology, with profound implications both fundamental and translational [1]. Single cell technologies avoid the confounding factors emerging from averaging over potentially heterogeneous cell populations [2], providing a global map of biological cell-to-cell variability at the molecular level [3].

While single-cell transcriptomic technologies are rapidly reaching maturity, more recent platforms have emerged that enable simultaneous large scale measurements of multiple molecular layers within the same cell. Multi-omics assays can now capture DNA methylation and gene expression [4, 5], gene expression and copy-number variations [5], DNA accessibility and gene expression [6, 7], and chromatin accessibility along with DNA methylation and gene expression [8] for the same cell. Such platforms have enormous potential to elucidate the mechanisms of epigenetic regulation in unprecedented detail.

Despite the huge potential for breakthroughs, technical limitations in multi-omics technologies create formidable statistical challenges in the interpretations of their results. Single-cell sequencing technologies are notoriously affected by high noise levels, including very strong data sparsity. Such problems are amplified in multi-omics studies, where multiple independent sources of noise might affect the joint distribution of the measurements. Additionally, challenges with normalization strategies, batch effects or other latent variables related to cellular processes might further prevent biological components to emerge clearly from data [9]. As a result, direct adoption of classical statistical tools to assess associations between different molecular layers (e.g. Pearson or Spearman correlation) routinely leads to underpowered analyses, which are only able to identify a handful of significant associations [4, 8, 10].

In this paper, we argue that proper treatment of noise is essential in order to robustly retrieve significant statistical associations. To do so, we introduce SCRaPL (**S**ingle **C**ell **R**egulatory **P**attern **L**earning), a Bayesian hierarchical model to infer associations between different omics components. The Bayesian hierarchical framework, which has already been extensively used in single-omics single-cell analyses (e.g. [11, 12]), explicitly and transparently decomposes noise in the data, enabling efficient extraction of biological signals from technical noise. We demonstrate on both synthetic and real data sets that SCRaPL is both highly accurate and sensitive, identifying much larger numbers of statistically significant associations than standard correlation analyses while retaining a good control on false positives.

## Results

### SCRaPL: Single Cell Regulatory Pattern Learning

SCRaPL is a tool for exploratory analysis of high-throughput, single cell data, which aims to establish more robust associations between different molecular layers. The example we will focus on here is relating the expression of a specific gene with its epigenetic state, measured either by DNA methylation or chromatin accessibility, however different types of associations might be considered, by introducing alternative noise models within our framework.

Our starting point is the observation that correlations, while invaluable tools to generate hypotheses, are critically sensitive to noise. In particular, it is well known that adding uncorrelated noise to correlated random variables reduces the estimates of correlation, thus weakening statistical power in any analysis. To obviate this problem, SCRaPL introduces a hierarchical model, schematically described in Fig 1.

Briefly, for each cell $i$, the SCRaPL model associates observed values $\mathbf{Y_{ij}} = (Y_{ij1}, Y_{ij2})'$ for each feature $j$ (e.g., gene/ promoter pair) with a bivariate Gaussian vector (denoted as $\mathbf{X_{ij}} =$

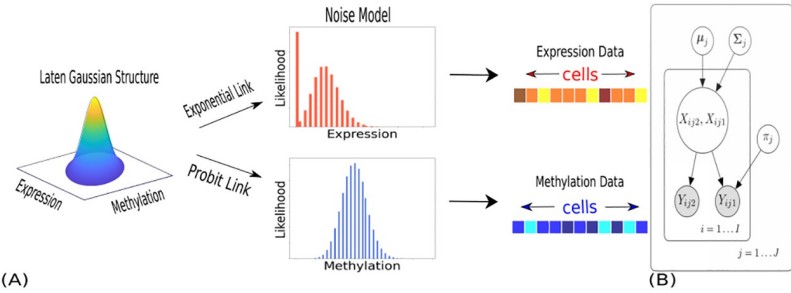

**Fig 1. Schematic and graphical representations of SCRaPL.** Here, we assume observed data consists of RNA expression and DNA methylation. 1A Schematic representation of the SCRaPL model. 1B SCRaPL's graphical model, depicting the statistical dependencies between observed genomic data ($Y_{ij1}$ is RNA expression; $Y_{ij2}$ is DNA methylation), their associated latent variables ($X_{ij1}$, $X_{ij2}$) and feature-specific model parameters ($\boldsymbol{\mu}_j$, $\Sigma_j$). The additional parameter $\pi_j$ is specific to the noise model that is assigned to RNA expression data and captures zero inflation. Full details are given in the model description section in *Methods*.

$(X_{ij1}, X_{ij2})')$ with unknown latent mean $\boldsymbol{\mu}_j$ and correlation matrix $\Sigma_j$. The latter is parameterized such that $\rho_j$ captures the feature-specific underlying correlation across both molecular layers. The latent variables $\mathbf{X_{ij}}$ are then passed through a suitable nonlinear link function to generate the expected value of the observation. The observation noise model, as well as the nonlinear link function, are tailored to the type of assay being analysed (and can also be designed in a data-driven fashion by using model selection techniques). In particular, we use a zero inflated Poisson noise model for RNA expression and binomial noise models for DNA methylation or chromatin accessibility; full details are given in the model description subsection, Eqs (1)–(4), *Methods* section. SCRaPL then uses Bayesian inference to reconstruct the latent mean values and correlation $\rho_j$ from independent observations over many cells. A probabilistic decision rule together with Bayesian multiple testing correction methods [Expected False Discovery Rate, EFDF; [13]] can be deployed to quantify association strength and associate statistical significance to the reported correlations.

## Benchmarking SCRaPL using synthetic data

To assess the estimation performance of SCRaPL, we experimented on synthetic datasets consisting of 300 simulated features (pairs of gene expression and promoter methylation values). The experiments were varied to cover a number of different scenarios: numbers of cells; coverage levels; fraction of zeros in expression data (zero inflation, ZI); as well as different latent mean and covariance structures. A detailed description of the various simulation scenarios, is provided in Table 1 and S3 Text.

Here, we primarily focus on estimation accuracy for the feature-specific latent correlation $\rho_j$ but also summarize results for other parameters to get the complete view. Violin plots summarizing the difference of SCRaPL's posterior from generating parameters as a function of cells can be found in Fig 2. Results for other model parameters are displayed in S3 Text (See A-I Figs in S3 Text).

We start by considering a situation of perfect model specification (experiments 1–3 in S3 Text), in order to assess the identifiability of our model and to document the degradation of correlation estimates obtained with classical methods. In this case, we observe that all methods provide estimates of correlation with zero-mean expected error, with an accuracy which increases with the number of cells in the data set. However, particularly for relatively low numbers of cells, the accuracy of the estimates was considerably higher for SCRaPL than for the

**Table 1. Summary of synthetic data experiments.** In all cases, latent means and standard deviations were set as $\mu_{j1}$ = 4, $\mu_{j2}$ = 1, $\sigma_{j1}$ = 3 and $\sigma_{j2}$ = 2. Unless otherwise stated, our simulations were based on: $I$ = 60 cells, $J$ = 300 features, 20% ZI rate on average for the expression data ($\pi_j$ = 0.20) and an average methylation coverage ($n_{ij}$) equal to 275 (sampled from a Uniform distribution with range [50, 500]) across cells and genes. When varying the number of cells, we use $I \in$ {5, 10, 25, 50, 100, 200, 400, 800, 1600}. When varying expression ZI, we use $\pi_j \in$ {0.1, 0.2, 0.3, 0.4, 0.5, 0.7, 0.8}. When varying methylation coverage, we sample $n_{ij}$ from Uniform distributions with ranges given by [5, 10], [10, 20], [20, 50], [50, 250] and [500, 1000]. Full details are provided in S3 Text.

| Experiment | Description |
|---|---|
| 1 | Correlations $\rho_j$ sampled from a Beta(15, 15) distribution, varying number of cells. |
| 2 | Correlations $\rho_j$ sampled from a Beta(15, 15) distribution, varying expression ZI. |
| 3 | Correlations $\rho_j$ sampled from a Beta(15, 15) distribution, varying methylation coverage. |
| 4 | Correlations $\rho_j$ sampled from a $U[-0.8, -0.6]$ distribution, varying number of cells. |
| 5 | Correlations $\rho_j$ sampled from a $U[-0.8, -0.6]$ distribution, varying expression ZI. |
| 6 | Correlations $\rho_j$ sampled from a $U[-0.8, -0.6]$ distribution, varying methylation coverage. |
| 7 | As experiment 1, but latent expression means sampled from scVI. |
| 8 | As experiment 2, but latent expression means sampled from scVI. |
| 9 | As experiment 3, but latent expression means sampled from scVI. |

classical Pearson and Spearman methods. Fig 2A shows a comparison of the three methods as we vary the number of cases, with a ZI level fixed to 20% (a benign setting similar to what encountered in high-depth plate-based technologies). SCRaPL outperforms both Spearman and Pearson by a large margin for all numbers of cells considered. Even in the most favourable case of 1600 cells (an unreasonably large number of cells for plate-based technologies), Pearson and Spearman systematically underestimate the (absolute value) of the correlation, while SCRaPL returns an accurate estimation for all true correlation values, as shown in the scatter-plot in Fig 2B. So, while overall all methods are unbiased in their estimates, Spearman and Pearson systematically underestimate the absolute value of the correlation, potentially leading to lesser power (see next section). As expected, the performance for all methods degrades with increasing levels of ZI (see S3 Text). However, we did not observe significant differences for SCRaPL correlation estimates across different levels of coverage (see S3 Text).

To probe the importance of prior specification, we generated data where the underlying correlation values $\rho_j$ were in an area with low prior mass (experiments 4, 5 and 6 in S3 Text). In this case, we did observe some bias in our estimates (see Figs D-F in S3 Text), particularly

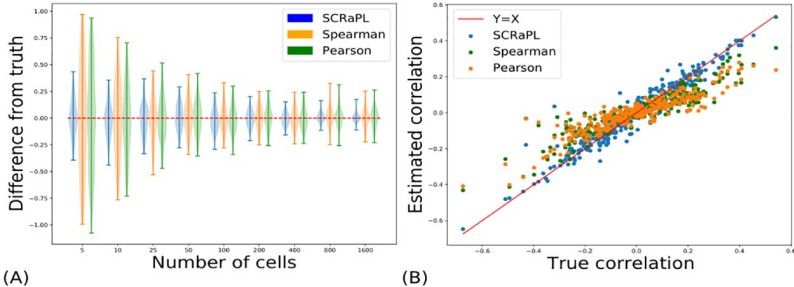

(A)　　　　　　　　　　　　　　　　(B)

**Fig 2. Plots summarizing differences in correlation estimation between SCRaPL, Spearman in Experiment 1 with synthetic data.** (2A) Estimated correlation difference from true correlation as a function of cells for SCRaPL, Spearman and Pearson. (2B) Estimated correlation as a function of true correlation for SCRaPL, Spearman and Pearson in synthetic datasets with 300 genes and 1600 cells. Each dot represents a gene and is color-coded based inference approach.

when the number of cells is low. Similarly, performance diminishes with increasing ZI levels and stays relatively intact across different coverage levels.

As a final test of more severe model mismatch, we evaluated predictive performance in a scenario where we retained the same noise model, but replaced the latent multivariate Gaussian distribution by expression rates inferred using a variational auto-encoder [scVI; [14]] that was trained on the scRNAseq data from [15] (see S3 Text, experiments 7–9). Despite the model mismatch, we observed good estimation performance for $\rho_j$ across a range of simulation parameters (see Figs G-I in S3 Text).

## SCRaPL improves the power to identify associations between molecular layers in mouse embryonic stem and brain cells

We next consider two single cell multi-omics datasets generated by scNMT-seq [8] and the 10x Genomics Multiome ATAC plus Gene Expression platform. Samples correspond to mouse embryonics stem cells (mESC) and brain cells (fresh cortex, hippocampus, and ventricular zone) (mEBC) at various developmental stages (embryonic days 4.5, 5.5, 6.5,7.5 for mESC and 18 for mEBC), which comprise the exit from pluripotency and primary germ layer [15] and the end of retinal ganglion cell generation [16].

In mESC cells we are investigating correlations between methylation for protein coding promoters within ±2.5kbps from Transcription Start Site (TSS) and expression. The mEBC data set consists instead of accessibility (measured by ATAC seq) and expression data. In that case we quantify the associations between expression and accessibility of enhancers lying in a region of 12.5kbps form the gene (an analysis of associations between expression and promoter accessibility for mESC cells is shown in S3 Text). Importantly, the two data sets were obtained using technologies with widely differing technical characteristics: the plate-based scNMT platform returns good coverage levels (and hence low ZI) for a limited number of cells, while the 10x platform assays many more cells but with lower coverage and higher dropout rates. After quality control, the resulting data sets contained 9480 features (gene promoters) and 679 mESCs and 4249 features (enhancers) and 4052 mEBCs respectively (Methods).

To compare the power of different methods to detect associations between molecular layers, we considered, alongside SCRaPL, the classical Spearman and Pearson correlation tests (Methods). The latter in particular has been widely used for single cell multi-omics data (e.g. [4, 8]); neither method takes into account noise in producing estimates of correlation. Molecular layer associations were retrieved as significant by controlling EFDR and FDR to 10%, respectively.

Fig 3 shows the summary results of these analyses. On the ESC data set, SCRaPL retrieves approximately 2.5 times more associations compared to both Pearson and Spearman testing, retrieving 217 (SCRaPL) versus 68(Pearson)/85(Spearman) (Fig 3C and Table A in S4 Text) associations. Fig 3A shows Bayesian Volcano plots, demonstrating how SCRaPL captures many more associations than frequentist alternatives. The overwhelming majorities of the associations recovered by frequentist methods are also captured by SCRaPL (Pearson and Spearman tend to be in very good agreement on this data set), which captures many more associations. We also looked at accessibility-expression pairs but due to weak signal no significant features were found by any of the methods. Later, we will investigate the biological significance of these results, showing how the greater statistical power of SCRaPL does in fact afford greater insights in the underlying biology.

Fig 3B and 3D show the analogous results for the analysis of EBC data. Here the picture is completely different: while SCRaPL still detects many more associations than Pearson, Spearman testing collapses and can only detect one significant associations. This points to a statistical vulnerability of Spearman testing when applied to data with high zero inflation in both

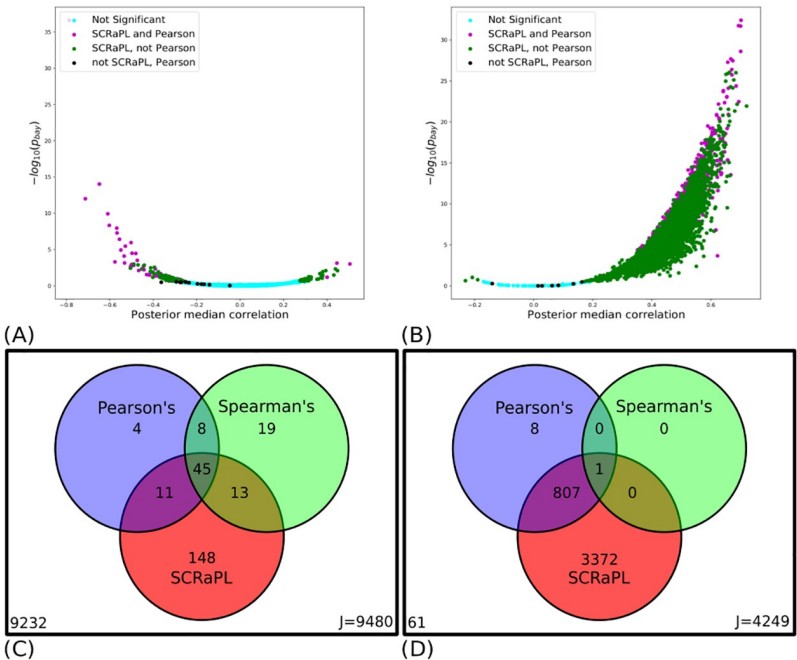

**Fig 3. Summary of experiments on real data.** Figures summarizing most important points from synthetic and real data experiments. (3A, 3B) Bayesian volcano plots for mESC and mEBC data respectively. Scatter plot of posterior probability under the null hypothesis (in log scale) as a function of posterior median correlation. Each dot represents a feature and is marked with different color depending the method that labels it as a significant association. (3C, 3D) Venn diagrams summarizing detection rates for SCRaPL, Pearson and Spearman in mESC and mEBC data. By accounting for different sources of noise it detects a large set of features identified by frequentist alternatives. SCRaPL also uncovers a additional large set that would be impossible for frequentist methods to identify in a robust way.

molecular layers. More precisely, the large number of zeros present in both expression and accessibility mEBC data creates a large set of ranking ties, creating an intrinsic mathematical problem for Spearman correlation. That is reflected in the 4180, 816 and 1 detected associations for SCRaPL, Pearson and Spearman (Fig 3D and Table B in S4 Text). Unsurprisingly, both SCRaPL and, to a lesser extent, Pearson testing identify as significant a greater fraction of association pairs between accessibility and expression, as expected from the basic biology of gene expression. In particular, the overwhelming majority of correlations between proximal enhancers accessibility and gene expression were deemed to be significant by SCRaPL, reflecting the importance of proximal enhancers in the regulation of gene expression.

While the ability of SCRaPL to detect larger numbers of associations is certainly an encouraging feature, it is essential to characterize whether this is due to greater power, or simply to a greater vulnerability to false positives. However, determining empirically the false positive rate is challenging as access to ground truth correlation values for each feature is impossible.

To address these issues, we proceed pragmatically by constructing negative control data sets in which observations of methylation and expression values for a particular feature in different cells are randomly permuted. This will destroy any correlation structure between the two quantities, so that features detected as significant in negative control data can be considered as false positives. Here, we constructed 5 negative control datasets. For all negative controls, SCRaPL and Pearson/Spearman testing only detected a handful of associations, consistently less than for the original data (see Tables A-B in S4 Text). These results suggest that all methods control for false positives, reinforcing the significance of the associations retrieved.

In summary, these results demonstrate that SCRaPL displays significantly increased statistical power in detecting associations between different molecular layers for both main types of multi-omics technological platforms. Intriguingly, both SCRaPL and Pearson testing appear to be largely insensitive to the type of technology, with SCRaPL identifying between 2.5 and 5 times more associations. Instead, Spearman testing reveals an intrinsic weakness in dealing with high sparsity data, making it potentially unsuitable as a tool for 10x multi-omics data analysis.

## SCRaPL associations are influenced by data sparsity and are robust to outliers

Fig 3A–3D (and Figs A-C in S6 Text) show clearly that, while most Pearson/ Spearman associations are also detected by SCRaPL, there are still some discrepancies. It is therefore natural to wonder to what extent the signals detected by the alternative methods are different, and what factors influence the different outcomes, in particular the much greater detection power obtained by SCRaPL.

From the modeling perspective, there are two major differences: first, SCRaPL considers noise models which capture overdispersion and take into account coverage in the epigenomic data. This should make SCRaPL associations less vulnerable to outlier values (eg. genes with low average expression with one or two high readings) or to epigenomic measurements with low coverage. Secondly, SCRaPL includes zero inflation in its accessibility/expression model, and can therefore attribute to that component some measurements of zero expression should the evidence dictate so. In the rest of this section, we present some empirical evidence that indeed supports the presence of these benefits in our real data analysis.

We consider the set of associations which are called as significant by at least one method, and split it into 3 categories: agreement between predictions, association labeling as significant by SCRaPL, but not by Pearson/Spearman testing, and vice-versa. We then analyze these three sets attempting to detect common patterns, discussing some examples to substantiate our findings.

Features for which Pearson/Spearman testing and SCRaPL agree tend to have high coverage and small number of zeros in case of expression (or accessibility in 10X). An example feature called as significant by SCRaPL and Pearson is in Fig 4A.

To gain more insight on the factors driving SCRaPL inferences it is interesting to focus on associations, whose significance differs between the two methods. An example of an association detected by Pearson/Spearman testing but not SCRaPL is shown in Fig 4B. As we can see, we have a large fraction of zero expression values with very low methylation coverage. As a result, SCRaPL, while placing most of the posterior mass over negative correlation values, cannot confidently exclude the possibility of no correlation. This example perfectly illustrates that divergences between SCRaPL and Pearson/Spearman testing are often driven not by expected values, but by the fact that SCRaPL additionally performs uncertainty quantification on its results.

An example of an association deemed significant by SCRaPL, but not by Pearson/Spearman testing, is shown in Fig 4C. In this case, we tend to have medium to high expression and good coverage. However, Pearson/Spearman correlation remain below detection levels due to a number of observations with zero expression. This is an example where SCRaPL can be particularly beneficial, since the noise model can better capture potential effect of zero inflation.

To provide a more quantitative, global explanation of the differences between SCRaPL and Pearson, we regress the absolute difference in inferred correlation against methylation coverage and percentage of zero counts for each feature across all cells. The resulting regressions,

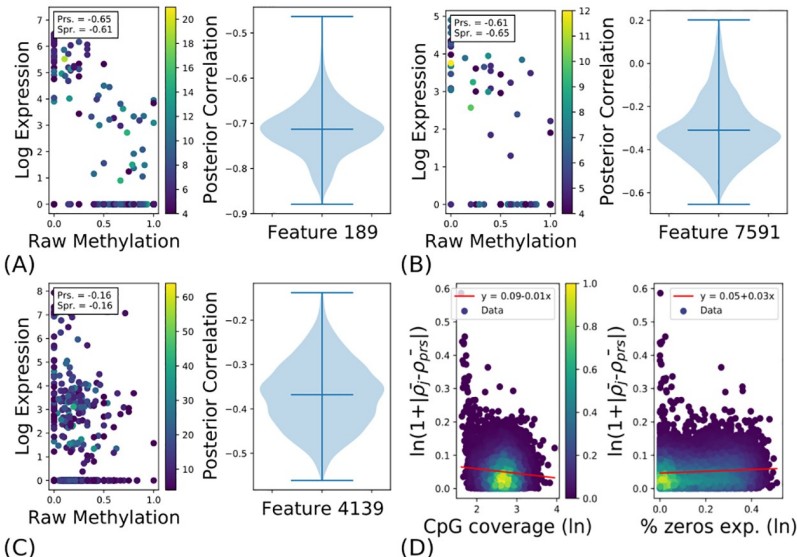

**Fig 4. SCRaPL's behavior compared to Pearson/Spearman correlation in micro and macro scale.** In all figures apart from 4D the scatter plot depicts raw data for chosen features color-coded by CpG coverage, and normalized expression plotted in the $log(1 + x)$ scale. The violin plots depict the posterior correlation densities estimated by SCRaPL for the raw data in their left hand side. (4A) Example where both SCRaPL and Pearson/Spearman identify the feature's association as significant. (4B) Example were only Pearson/Spearman identifies the feature's association significant. (4C) Example were only SCRaPL identifies the feature's association significant. (4D) Scatter plots to demonstrate the negative/positive relationship between alternative correlation estimates and CpG coverage/% zeros in expression respectively. ($\bar{\rho}_j$ and $\rho_{prs}$ in Fig 4D are posterior mean and Pearson correlation for feature $j$.).

shown in Fig 4D, demonstrate a weak but consistent effect of both forms on noise, confirming that differences between the two methods are more prominent in noisier situations where methylation coverage is low or sparsity is high. This analysis is also confirmed for Pearson and Spearman in both mESC and mEBC data in A Fig of S11 Text.

## SCRaPL identifies biologically meaningful epigenetic regulation in early mouse gastrulation

To provide further biological support to SCRaPL associations, we perform our own exploratory analysis in early gastrulation phases using SCRaPL significant findings.

We start by choosing early pluripotency and germ cell markers where methylation's strong repressive role is widely investigated (e.g. [15, 17, 18]). Developmental pluripotency markers (ie. Dppa2,Dppa4,Dppa5a) exhibit strong regulatory patterns with the generally high expression levels in days 4.5 and 5.5 being gradually suppressed as cells diversity to progenitors of major organs. Methylation's strong silencing role was also found in Dnmt3l, a catalytically inactive DNA methyltransferase that cooperates with Dnmt3a and Dnmt3b to methylate DNA [19]. In addition, our analysis identified a series of genes with strong regulatory action vital to embryonic development Atp6v0d1 [20], to spermatogenesis/placenta-supported development Tex19.1 [21] and others with unknown roles like Zfp981 and Trap1a.

To complete the exploratory analysis, we look at Gene Set Enrichment Analysis (GSEA) using DAVID [22] to establish links with biological phenomena observed in early embryogenesis and gene promoter methylation. To identify the processes we allow a minimum of 7 genes, a p-values up to 0.3 and sort them based on their enrichment score. As a result we have identified a total of fifteen developmental and house-keeping processes (see Fig A in S7 Text). The

highest enrichment scores are encountered for angiogenesis and in utero embryonic development with 2.6 and 2.1 respectively. For house-keeping processes we get proteolysis, ion transport and negative regulation of transcription with enrichment scores 2.2,1.9 and 1.7 respectively. Using the same filtering parameters in DAVID with the set of genes detected by Pearson we would recover a single process, regulation of transcription with enrichment score 1.5. Spearman testing detects a larger number of associations than Pearson on the ESC data set, and consequently has increased power in detecting enriched processes. In this case the number of recovered processes increases to 7, which consist however of primarily house-keeping processes (see Fig B in S7 Text).

This analysis confirms the biological plausibility of the identified SCRaPL associations. It should be emphasised that the enrichment analysis has only been possible due to the larger number of associations identified by SCRaPL: GSEA analyses require considerable numbers of genes to identify any significant enrichment. This underlines the fact that technical variability not only erodes correlation but significantly under-powers downstream exploratory analysis in multi-omics data. Hence by modelling data generative processes we can increase substantially the scope of downstream interpretative analyses of single-cell multi-omics data.

## Using SCRaPL as a data denoising tool

The detection of associations between layers is only one of the many possible analyses which can be performed on multi-omics data sets. A substantial line of research has recently emerged around the topic of data integration, which aims to combine data from multiple layers measured in different cells obtained from the same biological system. The goal of such analyses is to enhance our understanding of cellular identity and function [23]. Popular platforms like Seurat [24] implement data integration via a dimensionality reduction approach based on Canonical Correlation Analysis (CCA), a technique based on Singular Value Decomposition of empirical correlation matrices [25]. Despite its proven capabilities, CCA is not designed to handle count data. We therefore wondered whether SCRaPL's likelihoods tailored on specific data formats could under certain cases provide a valuable addition to the integration pipeline. Specifically, here we use SCRaPL as a denoising tool, and perform data integration at the level of the latent variables, rather than the raw data. In this subsection we follow the vignettes provided by Seurat's authors [26] and compare the results with and without SCRaPL's denoising. We note that this analysis is only a proof of concept as SCRaPL uses multi-omics data collected in the same cells as input and therefore such integration is not required.

For comparison between SCRaPL denoised and raw data we looked at peripheral blood mononuclear cells (PBMC) data [27]. This dataset contains expression and accessibility for 12000 PBMCs gathered from a healthy 25 year old donor, see Methods for details on data preprocessing.

To perform data integration, we remove cell specific noise by sampling latent space accessibility/expression from the respective posterior distributions obtained from SCRaPL. In cases of peaks mapped to multiple genes, readings were averaged. These data were integrated by Seurat [24], ignoring TF-IDF(Seurat's accessibility data preprocessing) and scRNA normalization (aimed at expression data) steps. Standard performance monitoring plots like label transfer between single cell expression and accessibility data as well as integration plots are presented in Fig 5. In general epigenomic and transcriptomic layers have integrated well for both raw and SCRaPL preprocessed data as suggested from Fig 5C and 5D. This picture remains consistent across multiple other trials (see Fig A in S12 Text). The integration metrics found in Fig 5A and 5D show a comparable performance between SCRaPL preprocessing and raw data. This is in stark contrast with the results of the preceding sections, which

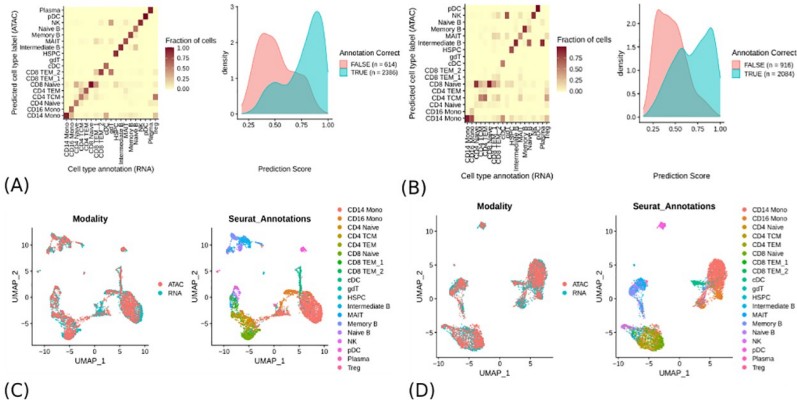

**Fig 5. Cell label transfer from expression to accessibility data for raw 5A and SCRaPL 5B preprocessed data.**
Visualization of sc-RNA and scATAC data on the same plot for raw 5C and SCRaPL 5D preprocessed data.

demonstrated a consistent superiority of SCRaPL in detecting associations at the level of individual features. The reason for this is probably to be found in the dimensionality reduction performed by Seurat: canonical components found by CCA are obtained via an averaging process which already does an excellent job at filtering out noise, much in the way that robust principal components can often be extracted also from noisy data.

## Discussion

Single cell multi-omics sequencing technologies are rapidly becoming an important tool to understand epigenetic regulation for individual cells in complex biological processes, such as early embryo development. However, analysis of such data still presents a major bottleneck, due to the high-dimensionality, sparsity and heterogeneous noise affecting them. In this paper, we argued that the introduction of noise-aware approaches is fundamental in developing the field of single-cell multi-omics. We introduced SCRaPL, a Bayesian approach to perform perhaps the most basic and common multi-omics analysis, the discovery of correlative associations between different data modalities. By employing dedicated noise models in a latent-Gaussian framework, SCRaPL achieves more powerful and more robust results than simple analyses based on Pearson correlation, which is by far the most widespread tool currently used.

Our analyses were based on existing annotation, where the expression of a given gene was correlated with epigenetic data from a nearby genomic region (promoters or nearby enhancers). This appears to be a reasonable demonstration of the tool, although it clearly limits the scope for discovery of interesting biological processes such as distal regulation. It should be pointed out that SCRaPL could also be used to test associations between unannotated regions along the lines explored in e.g. [12], [28].

The Bayesian hierarchical framework employed by SCRaPL also offers a template for the application of more complex analysis techniques (such as clustering, dimensionality reduction and network inference) to multi-omics data. In many analyses, we expect that consistent handling of noise will be valuable, although it should be pointed out that some downstream analyses already perform noise filtering implicitly. This was demonstrated in our comparison with the CCA approach implemented in Seurat [24], which effectively averages out noise during dimensionality reduction, yielding very similar results to SCRaPL. As with most Bayesian methods, SCRaPL does suffer from a higher computational burden, particularly when

compared with extremely simple analyses, such as Pearson correlation. Extension of noise-aware Bayesian methods to different single-cell multi-omics analyses may therefore require the adoption and evaluation of more efficient computational inference techniques, such as variational inference [29].

## Materials and methods

### A Bayesian hierarchical framework for noisy single cell multi-omics data

SCRaPL implements a Bayesian hierarchical approach that is tailored to the data generated by single cell multi-omic assays. Here, we assume that matched data is available for two molecular phenotypes, but our formulation is flexible and can in principle be expanded to include additional layers. A graphical representation for the model implemented in SCRaPL is provided in Fig 1. The distribution of a latent vector $\mathbf{X}_{ij}$ is used to capture the association across molecular layers. For each cell $i$ ($\in\{1, \ldots I\}$) and feature $j$ ($\in\{1, \ldots J\}$), the latter is given by

$$\mathbf{X}_{ij} = \begin{pmatrix} X_{ij1} \\ X_{ij2} \end{pmatrix} | \boldsymbol{\mu}_j, \boldsymbol{\Sigma}_j \overset{ind}{\sim} \mathrm{N}(\boldsymbol{\mu}_j, \boldsymbol{\Sigma}_j),$$ (1)

where

$$\boldsymbol{\mu}_j = \begin{pmatrix} \mu_{j1} \\ \mu_{j2} \end{pmatrix} \text{ and } \boldsymbol{\Sigma}_j = \begin{pmatrix} \sigma_{j1}^2 & \rho_j \sigma_{j1} \sigma_{j2} \\ \rho_j \sigma_{j1} \sigma_{j2} & \sigma_{j2}^2 \end{pmatrix}.$$ (2)

In this formulation, we assume independence across all features, which will be analyzed separately (this enables trivial parallelization across features). Different noise models are then assigned to each molecular layer based on the properties of the associated data. There are two different likelihoods that we use depending the types of cells we use. For count data (i.e. gene expression in mESC/mEBC and chromatin accessibility in mEBC) we use a zero-inflated Poisson noise and for the rest (i.e. DNAm and accessibility in mESC) we use a Binomial distribution. Specific noise models for each of the data types considered here are described below.

**RNA expression noise model.** Let $Y_{ij1}$ be a random variable representing the number of raw read-counts observed for each cell $i$ and feature $j$. Conditional on the value of the latent variable $X_{ij1}$, we use an exponential link function and assume that

$$\mathbb{P}\Big(Y_{ij1} = y_{ij1} | X_{ij1} = x_{ij1}, s_i, \pi_j\Big) = \begin{cases} \left(1 - \pi_j\right) \dfrac{(s_i \mathrm{e}^{x_{ij1}})^{y_{ij1}} \, \exp(-s_i \mathrm{e}^{x_{ij1}})}{y_{ij1}!} & \textbf{if } y_{ij1} > 0, \\ \pi_j + (1 - \pi_j) \, \exp\left(-s_i \mathrm{e}^{x_{ij1}}\right) & \textbf{if } y_{ij1} = 0. \end{cases}$$ (3)

The latter corresponds to a zero-inflated Poisson (ZIP) model with an exponential link, where $s_i$ ($> 0$) is a cell-specific scaling factor that accounts for global differences across cells (e.g. due to sequencing depth) and $\pi_j$ ($\in[0, 1]$) represents a zero-inflation parameter (if $\pi_j = 0$, Eq (3) reduces to a Poisson model). The exponential link function leads to a zero-inflated Poisson-lognormal model, whose variations have been previously used for single cell RNA sequencing (scRNAseq) data [30, 31]. In practice, we infer scaling factors $s_i$ using scran [32] and use them as known model offsets.

The need for a zero-inflation component is a matter of debate for scRNA-seq data [33] and may depend on the experimental protocol used to generate the data. See *Comparing between alternative models* later in this section for a quantitative approach to evaluate the need for zero-inflation in specific datasets.

**DNAm noise model.** For each cell $i$ and feature $j$, let $n_{ij}$ be the number of CpG sites within a pre-specified genomic region (e.g. gene promoter) for which DNAm reads were obtained. These capture differences in coverage across cells and features. The conditional model for the number of methylated CpG sites $Y_{ij2}$ is then assumed to follow a binomial distribution such that

$$\mathbb{P}\Big(Y_{ij2} = y_{ij2}|X_{ij2} = x_{ij2}, n_{ij}\Big) = \binom{n_{ij}}{y_{ij2}}(\Phi(x_{ij2}))^{y_{ij2}}(1 - \Phi(x_{ij2}))^{n_{ij}-y_{ij2}}, \tag{4}$$

where $\Phi(\cdot)$ denotes a probit link function.

**Chromatin accessibility noise model.** The choice of noise model depends on the format of the input data. If the data consists of the number of features $Y_{ij2}$ in a genomic region [e.g. as in [34]], our modelling approach analysis follow the one described for RNA expression data. If the data consists of open peaks within a genomic region (e.g. as for the 10X scATAC seq protocol), the same binomial noise model used for DNA methylation data can be applied.

**Parameter interpretation.** To aid the interpretation of each model parameter, mean and variance expressions are derived for the noise models introduced above after integrating out the distribution of the latent vector $\mathbf{X}_{ij}$ (see S8 Text). In both cases, $\mu_{j1}$ and $\mu_{j2}$ control the overall RNA expression and DNAm values for the population of cells under study. Moreover, $\sigma_{j1}$ and $\sigma_{j2}$ capture the excess of variability (overdispersion) that is observed with respect to the baseline noise model. Finally, $\rho_j$ captures the latent correlation between molecular layers.

## Prior specification

A popular prior choice for covariance matrices is the inverse Wishart distribution. However, this has been shown to bias correlation coefficients depending whether marginal variances are small or large [35]. Instead, [36] used a separation strategy to decouple correlation from marginal variances. Our prior specification for $\Sigma_j$ is based on the parametrization introduced in Eq (2), with independent priors assigned to all feature-specific parameters. Our prior specification is given by

$$\pi_j \overset{ind}{\sim} \text{Beta}(a_j, b_j), \tag{5}$$

$$\boldsymbol{\mu}_j \overset{iid}{\sim} \mathbf{N}(\mathbf{m}, \mathbf{H}), \tag{6}$$

$$\sigma_{j1}, \sigma_{j2} \overset{iid}{\sim} \text{Inv-Gamma}(c_1, c_2), \tag{7}$$

$$\rho_j \overset{iid}{\sim} \text{Beta}_{[-1,1]}(d_1, d_2). \tag{8}$$

In Eq (8), the prior for $\rho_j$ corresponds to a four-parameter Beta distribution, whose support has been scaled to be $[-1, 1]$. In order to avoid systematically favoring positive or negative correlations, we centered the prior at 0 by setting $d_1 = d_2$. Then we tuned these prior hyperparameters on negative control data (see S2 Text), eventually choosing Beta(15, 15) as it helped to suppress false positive detection rates. More information about this provided in S5 Text. For the remaining hyper-parameter values, default values were set as $a_j = 2$, $b_j = 8$ to encourage low zero inflation. Moreover, we set $c_1 = 2.5$, $c_2 = 4.5$ but keeping the parameters within a reasonable range will also work, $\boldsymbol{\mu}_j = (4, 0)'$ for mESC data, $\boldsymbol{\mu}_j = (4, 3)'$ for mEBC data, $\mathbf{H}$ was set to be a $2 \times 2$ identity matrix.

## Implementation

As the posterior distribution associated to the model above does not have a closed analytical form, inference is implemented using No-U-Turn Sampler [37], a state of the art variation of Hamiltonian Monte Carlo [38].

For all the analyses shown in this article, we obtained 5000 samples from this algorithm and discarded the first 3000 iterations (burn-in) before estimating model parameters. Parameters were optimized during burn-in to an acceptance ratio of 0.65. Convergence is monitored using the Gelman-Rubin criterion [39].

## A probabilistic rule to detect statistically significant associations across layers

SCRaPL identifies features with statistically significant correlation across multi-omics layers (e.g. RNA expression and promoter DNAm) based on the posterior distribution of feature-specific latent correlation parameters $\rho_j$. Our decision rule depends on whether the posterior mass for $|\rho_j|$ is concentrated around high values. As in [40], this is quantified by the following tail posterior probabilities

$$p_j(\gamma) = \mathbb{P}(|\rho_j| \geq \gamma), \tag{9}$$

where $\gamma\ (> 0)$ denotes a minimum correlation threshold. If $p_j(\gamma)$ is greater than a probability threshold $\alpha$, a statistically significant correlation is reported for feature $j$.

Suitable values for $\gamma$ and $\alpha$ could be chosen using different approaches. In principle, $\gamma$ can be fixed a priori by the user. Instead, we adopt a data-driven approach based on the distribution of feature-specific posterior estimates obtained for $|\rho_j|$ using negative control datasets (see S4 Text). Such distribution can be used to quantify the strength of correlation estimates that can be expected by chance for a given sample size and sequencing depth. As a default choice, we select $\gamma$ to match the 90% quantile of the distribution described above. For a fixed value of $\gamma$, a grid search is used to select $\alpha$ according to a target EFDR. The latter is defined as

$$\text{EFDR}_\alpha = \frac{\sum_{j=1}^{J}(1 - p_j(\gamma))\,\mathbb{I}\,(p_j(\gamma) \geq \alpha)}{\sum_{j=1}^{J}\mathbb{I}\,(p_j(\gamma) \geq \alpha)}, \tag{10}$$

where $\mathbb{I}(A) = 1$ if $A$ is true, 0 otherwise. Our default target EFDR is equal to 10%.

## Current approach based on Pearson/Spearman correlation

To date, single cell multi-omics analyses have primarily used the Pearson/Spearman correlation coefficient $r$ to quantify associations between different types of molecular data [e.g. [4, 8]]. These estimates are directly derived from the observed data and do not assume a specific noise model. As the input for this calculation, gene expression data is typically normalised [e.g. using scran; [32]] and subsequently log-transformed after adding a pseudocount, while DNAm is normalised by coverage [note that the addition of a pseudocount is arbitrary and has been shown to distort variance estimates; [31]].

Based on these estimates, statistically significant correlations are selected by contrasting the hypotheses $H_0$: $\|r\| \leq u$ and $H_1$: $\|r\| \geq u$, for some threshold $u$. To control the False Discovery Rate (FDR) across features, the Benjamini-Hochberg correction [41] is typically used.

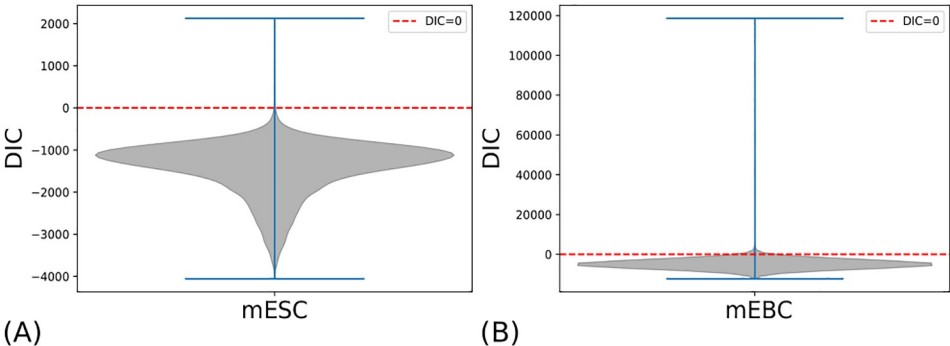

**Fig 6. DIC difference between model with and without inflation for mESC and mEBC data.** The more negative the difference, the stronger the evidence in favor of the model with zero inflation on the gene expression component and vice versa. As a visual reference, zero is marked with dashed red line.

## Comparing between alternative models

SCRaPL is a noise-aware approach with error models crafted to address challenges related to various multi-omics data. Our likelihood of choice for count data such as gene expression and accessibility in 10X data has been Poisson distribution. Since there is a debate surrounding the extend to which zero-inflation is required [33], we take an unbiased stance and use posterior model samples to perform model selection using Deviance Information Criterion [DIC; [42]]. DIC is a method for assessing goodness of fit while penalizing large effective numbers of parameters between alternative models, with lower DIC values indicating more preferable models.

To assess the need of zero-inflation in SCRaPL, we fit the zero-inflated and the standard Poisson in the methylation/expression the mESC data and accessibility/expression of mEBC data. For the large majority of features in the mESC and mEBC, DIC favors zero inflation as it is indicated from Fig 6.

## Single cell multi-omic datasets

We applied SCRaPL in the context of two single cell multiomic datasets. First, we consider the mESC dataset generated by [15] using the scNMT-seq protocol [8]. For these data, our analysis focuses on the correlation between gene expression and DNAm. We also experimented with chromatin accessibility and gene expression pair with not much success due to sparsity in accessibility data. Our second case study considers mEBC data generated using the 10X Genomics Multiome ATAC plus gene expression platform. Quality control steps applied to both datasets are described in S1 Text.

To aggregate DNAm data from different mESC and link open DNA chromatin from mESC to nearby genes we follow a window based approach. Reads are mapped using the GRCm38 mouse genome (accession number GSE56879). For more information, the reader is directed to S1 Text. When looking at methylation/gene expression of promoter regions in mESC data, a window of ±2.5kbp was used. For chromatin accessibility/gene expression in the same dataset the window was ±0.25kbp. Similarly, for accessibility and expression in the mECB data we map enhancers to genes at most ±12.5 kbp away. To control how our window choices affect results, we experimented with multiple window sizes, noticing minimal impact on the results.

Subsequently, a quality control step was applied to both datasets. For mESC data we removed features with zero variance in each modality and for which the percentage of expression zeros was above 80%. This resulted in a dataset with 9480 features and 679 cells. Similarly,

for mEBC data we removed features with more than 80% of zeroes in accessibility or expression, leading to a dataset with 4249 features and 4052 cells.

The data denoising analysis was done using peripheral blood mononuclear cells (PBMCs). For illustration purposes, we downsampled the dataset to 3000 cells by keeping only the ones with the highest sum across peaks. Then peaks and genes were reduced from 180000 to 30000 and from 36000 to 10000 respectively, based on their variability. Then the to $60k$ features in association magnitude were used by SCRaPL.

## Supporting information

**S1 Text. Data preprocessing.** Here we discuss the preprocessing and quality control steps taken in mESC and bESC datasets. This includes aggregating raw epigenetic data from multiple cells, normalizing single cell transcriptomic data, integrating epigenomic and trascriptomic layers and removing low quality data.
(PDF)

**S2 Text. Creating negative control datasets.** In this section we describe step by step the generation of negative control data, explaining our address to problems like missing coverage.
(PDF)

**S3 Text. Synthetic data.** We include an extensive analysis of synthetic data experiments used to develop SCRaPL. In particular, we include three sets of experiments that investigate SCRaPL's performance as a function of cells, zero-inflation and coverage. The first set is performed on data sampled form the model, the second on data sampled from the model and the correlation from a $U[-0.8, -0.6]$, and the third partly sampled from a deep generative model and partly from the model.
(PDF)

**S4 Text. Negative control experiments.** In this section we lay the detection rate comparisons between SCRaPL and Pearson for methylation/expression of mESC, accessibility/expression of mESC and accessibility/expression of bESC.
(PDF)

**S5 Text. Choosing between correlation priors.** In this section we present a data driven approach for choosing prior hyper-parameters.
(PDF)

**S6 Text. Extended comparisons between SCRaPL,Pearson and Spearman predictions.** In this section a more extended comparison between SCRaPL, Pearson and SPearman predictions is provided.
(PDF)

**S7 Text. Gene set enrichment analysis.** The complete gene set enrichment analysis using DAVID is provided.
(PDF)

**S8 Text. Connecting SCRaPL error model to likelihoods currently employed by practitioners.** We demonstrate that SCRaPL's expression likelihood serves as valid alternative as it exhibits the over-dispersion property that practitioners seek.
(PDF)

**S9 Text. Null hypothesis testing.** We give a thorough description of the hypothesis testing done to identify regions with strong regulatory action.
(PDF)

**S10 Text. Efficiency analysis.** We provide evidence of SCRaPL's scaling as a function of problem size.
(PDF)

**S11 Text. Macroscopic analysis of SCRaPL behavior compared to Pearson and Spearman Correlation.** We provide additional figures to Fig 3D that demonstrate the different behavior that SCRaPL presents from Pearson/Spearman correlation when high zero inflation and low CpG coverage is present in data.
(PDF)

**S12 Text. Stability of denoising.** Figures summarizing scRNA and scATAC integration using posterior latent space samples.
(PDF)

## Acknowledgments

We would like to thank Chantriolnt-Andreas Kapourani, Michalis Michaelides for their valuable comments and discussion.

## Author Contributions

**Conceptualization:** Guido Sanguinetti.

**Data curation:** Christos Maniatis.

**Methodology:** Christos Maniatis.

**Software:** Christos Maniatis.

**Supervision:** Catalina A. Vallejos, Guido Sanguinetti.

**Validation:** Catalina A. Vallejos, Guido Sanguinetti.

**Visualization:** Christos Maniatis.

**Writing – original draft:** Christos Maniatis.

**Writing – review & editing:** Christos Maniatis, Catalina A. Vallejos, Guido Sanguinetti.

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
