## [Editor Report · Decision Letter 0]

27 Sep 2021

Dear Dr Maniatis,

Thank you very much for submitting your manuscript "SCRaPL: hierarchical Bayesian modelling of associations in single cell multi-omics data" (PCOMPBIOL-D-21-01603) for consideration at PLOS Computational Biology. As with all papers, your manuscript was reviewed by members of the editorial board. Based on our initial assessment, we regret that we will not be pursuing this manuscript for publication at PLOS Computational Biology. 

We found that the manuscript would require a significant amount of revision to reach the quality of formal submission. The current issues include the inconsistent notations, widespread typos (e.g., Figure 4 and its caption are hardly comprehensible), and insufficient real data evidence. We would like to see another real dataset where the proposed method shows significant advances. In addition to the Pearson correlation, comparison with existing single-cell methods such as scLink is also necessary to show the advantage of the proposed method. If you find these comments addressable, please submit your revised manuscript as a new submission. Please also fully address the comments of the three ReviewCommons reviewers.

We are sorry that we cannot be more positive on this occasion. We very much appreciate your wish to present your work in one of PLOS's Open Access publications. Thank you for your support, and we hope that you will consider PLOS Computational Biology for other submissions in the future.

Sincerely,

Jingyi Jessica Li 

Guest Editor

PLOS Computational Biology

Sushmita Roy

Deputy Editor

PLOS Computational Biology

---

## [Decision Letter · Decision Letter 1]

9 Feb 2022

Dear Dr Maniatis,

Thank you very much for submitting your manuscript "SCRaPL: hierarchical Bayesian modelling of associations in single cell multi-omics data" for consideration at PLOS Computational Biology.

As with all papers reviewed by the journal, your manuscript was reviewed by members of the editorial board and by several independent reviewers. In light of the reviews (below this email), we would like to invite the resubmission of a significantly-revised version that takes into account the reviewers' comments.

We cannot make any decision about publication until we have seen the revised manuscript and your response to the reviewers' comments. Your revised manuscript is also likely to be sent to reviewers for further evaluation.

Sincerely,

Jingyi Jessica Li

Guest Editor

PLOS Computational Biology

Sushmita Roy

Deputy Editor

PLOS Computational Biology

Reviewer's Responses to Questions

**Comments to the Authors:**

Reviewer #1: In this work, the authors developed a method to identify region-gene association based on single-cell multi-omics data. The method is based on a Bayesian hierarchical model, which uses zero-inflated Poisson model with logit link function for the modalities of gene expression and chromatin accessibility, and binomial model with probit link for the modality of methylation. Overall, the method and study look solid. This is also an important problem in single-cell multi-omics. This could be a nice addition to the literature. I think Spearman correlation is more commonly used under this setting, because of its robustness to outliers. It will be good to also include Spearman correlation in the comparison.

Reviewer #2: Maniatis et al proposes a methods named SCRaPL to investigate the correlation in multi-omics single-cell datasets. The method is novel in its usage of a Bayesian hierarchical model to infer associations between different omics components. If the claim that the method has higher power and a good control on false positives can be better supported in analysis, this will be a potentially useful method in single-cell studies.

Writing:

Since a significant amount of descriptions is included in the supplementary file, the authors need to improve the clarity of the main text to help readers navigate between the manuscript and the supp file. I found myself spending a lot of time searching for explanations in the supp file.

Methods:

It is not clear to me what’s the meaning of Y. From formula (3), it seems that it should represent raw counts. However, the supplementary methods mention that the RNA data is normalized during preprocessing. Please clarify.

The authors need to explain how the data (except for gene expression) is binarized in order to use the Binomial distribution. Would the binarization cutoff significantly impact the final results?

Is there any justification for the usage of the probit link function in formula (4)?

Key derivation steps to obtain the posterior distribution are not given. The distribution should be added to Method and the key steps should be included at least in the supp file.

No software package is available for others to use the method.

Results:

In the experiments with synthetic data, (1) what’s the definition of “gene coverage”? (2) I would suggest moving the plots of true and inferred correlations to the main manuscript. (3) The Method section describes the approach to identify statistically significant correlation using SCRaPL. Can the authors show the accuracy of this method on these datasets?

In the analysis of mESC data, “a dataset with 9480 features and 679 cells” was used. This number is much smaller than the possible number of features. How many genes or DNAm features are included in these 9480 features? How would it affect the performance of SCRaPL if a less stringent filtering is applied and more features are included? Similar questions apply to the mEBC data.

Can the authors also show the comparison between SCRaPL and Pearson’s correlation (power and false positive rate) using the aynthetic data?

The last Results section presents SCRaPL as a data denoising method, and performs Seurat integration with and without SCRaPL’s preprocessing. (1) From Figure 4, it is not clear to me that SCRaPL’s preprocessing improves the analysis. Can the authors provide some quantitative comparisons? (2) A more detailed description needs to be provided in Methods. With SCRaPL’s preprocessing, what data is provided as the input into Seurat? (3) Since the procedure involves sampling from posterior distributions, how different are the integration results if the data are sampled multiple times?

Reviewer #3: It seems that the author has largely addressed previous reviewers' comments. However, the authors need to check if every single comment has been replied. For example I don't see response for the first comment of the first reviewer. Also, the figure legends need to be improved to discuss each of the subplots. Such description is lacking for figures 2 and 4.

For the software package on Github, I don't see any instructions about how to use the software or how to reproduce the results in the paper. This needs to be significantly improved.

**Have the authors made all data and (if applicable) computational code underlying the findings in their manuscript fully available?**

Reviewer #1: Yes

Reviewer #2: None

Reviewer #3: Yes

PLOS authors have the option to publish the peer review history of their article (what does this mean?). If published, this will include your full peer review and any attached files.

Reviewer #1: No

Reviewer #2: No

Reviewer #3: No
---

## [Decision Letter · Decision Letter 2]

2 May 2022

Dear Dr Maniatis,

We are pleased to inform you that your manuscript 'SCRaPL: A Bayesian hierarchical framework for detecting technical associates in single cell multiomics data' has been provisionally accepted for publication in PLOS Computational Biology.

Best regards,

Jingyi Jessica Li

Guest Editor

PLOS Computational Biology

Sushmita Roy

Deputy Editor

PLOS Computational Biology

Reviewer's Responses to Questions

**Comments to the Authors:**

Reviewer #1: all my previous comments have been addressed.

Reviewer #2: The revised manuscript has addressed all my questions.

Reviewer #3: The authors have addressed all my concern.

**Have the authors made all data and (if applicable) computational code underlying the findings in their manuscript fully available?**

Reviewer #1: None

Reviewer #2: None

Reviewer #3: Yes

PLOS authors have the option to publish the peer review history of their article (what does this mean?). If published, this will include your full peer review and any attached files.

Reviewer #1: No

Reviewer #2: No

Reviewer #3: No

---

## [Editor Report · Acceptance letter]

13 Jun 2022

PCOMPBIOL-D-21-01603R2 

SCRaPL: A Bayesian hierarchical framework for detecting technical associates in single cell multiomics data

Dear Dr Maniatis,

I am pleased to inform you that your manuscript has been formally accepted for publication in PLOS Computational Biology. Your manuscript is now with our production department and you will be notified of the publication date in due course.

With kind regards,

Zsofia Freund
